# The conserved aspartate ring of MCU mediates MICU1 binding and regulation in the mitochondrial calcium uniporter complex

**Charles B Phillips[1], Chen-Wei Tsai[1,2], Ming-Feng Tsai[1,2]***

[1]Department of Biochemistry, Brandeis University, Waltham, United States; [2]Department of Physiology and Biophysics, University of Colorado Anschutz Medical Campus, Aurora, United States

**Abstract** The mitochondrial calcium uniporter is a $Ca^{2+}$ channel that regulates intracellular $Ca^{2+}$ signaling, oxidative phosphorylation, and apoptosis. It contains the pore-forming MCU protein, which possesses a DIME sequence thought to form a $Ca^{2+}$ selectivity filter, and also regulatory EMRE, MICU1, and MICU2 subunits. To properly carry out physiological functions, the uniporter must stay closed in resting conditions, becoming open only when stimulated by intracellular $Ca^{2+}$ signals. This $Ca^{2+}$-dependent activation, known to be mediated by MICU subunits, is not well understood. Here, we demonstrate that the DIME-aspartate mediates a $Ca^{2+}$-modulated electrostatic interaction with MICU1, forming an MICU1 contact interface with a nearby Ser residue at the cytoplasmic entrance of the MCU pore. A mutagenesis screen of MICU1 identifies two highly-conserved Arg residues that might contact the DIME-Asp. Perturbing MCU-MICU1 interactions elicits unregulated, constitutive $Ca^{2+}$ flux into mitochondria. These results indicate that MICU1 confers $Ca^{2+}$-dependent gating of the uniporter by blocking/unblocking MCU.
DOI: https://doi.org/10.7554/eLife.41112.001

*For correspondence:
ming-feng.tsai@ucdenver.edu

**Competing interests:** The authors declare that no competing interests exist.

## Introduction

The mitochondrial calcium uniporter is a multi-subunit $Ca^{2+}$-activated $Ca^{2+}$ channel complex located in the inner mitochondrial membrane (IMM). It catalyzes $Ca^{2+}$ influx from the intermembrane space (IMS) into the mitochondrial matrix, where a large quantity of $Ca^{2+}$ can be stored. Extensive studies have established that the uniporter regulates spatial and temporal dimensions of intracellular $Ca^{2+}$ signals, as well as $Ca^{2+}$-dependent mitochondrial processes, including oxidative phosphorylation and programmed cell death (*Kamer and Mootha, 2015*; *Rizzuto et al., 2012*).

The $Ca^{2+}$-conducting function of mammalian uniporters are mediated by two subunits, MCU and EMRE, in the transmembrane (TM) region (*Figure 1*). The MCU protein possesses two TM helices and a highly-conserved 'DIME' signature sequence (*Baughman et al., 2011*; *De Stefani et al., 2011*). High-resolution structures (*Baradaran et al., 2018*; *Fan et al., 2018*; *Nguyen et al., 2018*; *Yoo et al., 2018*) show that MCU assembles into a tetrameric $Ca^{2+}$ pore, with the DIME-Asp and -Glu forming two parallel side-chain carboxylate rings to constitute a $Ca^{2+}$ selectivity filter at the pore's IMS entrance (*Figure 1*). The single-pass EMRE protein binds to MCU via its TM helix (*Tsai et al., 2016*). This interaction is shown to be necessary for $Ca^{2+}$ permeation (*Tsai et al., 2016*; *Kovács-Bogdán et al., 2014*; *Sancak et al., 2013*).

The uniporter is tightly regulated by intracellular $Ca^{2+}$ signals. It stays quiescent in resting cellular conditions, and becomes activated only when IMS $Ca^{2+}$ increases to low micromolar levels (*Csordás et al., 2013*; *Mallilankaraman et al., 2012*). This $Ca^{2+}$-dependent gating is mediated by

two EF-hand (a helix-loop-helix $Ca^{2+}$-coordinating motif) containing subunits: MICU1 and MICU2 (the neuron-specific MICU3 is not discussed here) (*Csordás et al., 2013*; *Mallilankaraman et al., 2012*; *Perocchi et al., 2010*; *Plovanich et al., 2013*), which are tethered to the uniporter's TM region via the C-terminal tail of EMRE (*Tsai et al., 2016*). Depletion of MICU1 eliminates $Ca^{2+}$-regulation of the uniporter, causing the channel to constitutively load $Ca^{2+}$ into the matrix (*Tsai et al., 2016*; *Mallilankaraman et al., 2012*; *Plovanich et al., 2013*; *Tsai et al., 2017*), a condition linked to debilitating neuromuscular disorders in humans (*Logan et al., 2014*). Currently, the mechanism by which MICUs control $Ca^{2+}$ transport via MCU remains largely unknown.

Here, we demonstrate that MICU1 interacts with MCU's DIME-Asp via a $Ca^{2+}$-modulated electrostatic interaction. This is mediated by two closely-spaced Arg residues on the surface of MICU1. MICU2, which lacks these Args, does not bind MCU. Mutations that disrupt the MCU-MICU1 interaction severely perturbs $Ca^{2+}$-regulation of the uniporter. These results led to a molecular mechanism in which MICUs open or close the uniporter in response to intracellular $Ca^{2+}$ signals by physically blocking or unblocking the MCU pore.

## Results

### Evolutionarily conserved MCU-MICU1 interactions

Phylogenetic analyses (*Sancak et al., 2013*; *Bick et al., 2012*) have shown that uniporters in lower eukaryotes (*e.g.*, plants and protists) contain only MCU and MICU1 subunits, raising a possibility that MICU1 might gate MCU via direct molecular contacts. If so, these interactions might be conserved in evolution to ensure proper regulation of the uniporter. To test this idea, we performed co-immunoprecipitation (CoIP) experiments to examine complex formation between human MICU1 and various MCU homologues in MCU/EMRE-KO HEK 293 cells (*Tsai et al., 2016*). The EMRE gene is deleted because EMRE can bind both MCU and MICU1 (*Figure 1*) (*Tsai et al., 2016*; *Sancak et al., 2013*), and would therefore complicate assessment of direct MCU-MICU1 contacts. *Figure 2* shows that human MICU1 pulls down not only human MCU but also MCU homologues in *D. melanogaster*, *C. elegans*, *D. discoideum*, and *A. thaliana,* indicating that the MCU-MICU1 interaction is indeed evolutionarily conserved.

### The role of the DIME-Asp in $Ca^{2+}$ transport and MICU1 binding

We reasoned that MICU1 might bind to the DIME-Asp, as MCU structures (*Baradaran et al., 2018*; *Fan et al., 2018*; *Nguyen et al., 2018*; *Yoo et al., 2018*) show that this Asp is the only fully-conserved residue with the side-chain exposed to the IMS, where MICU1 is localized. Accordingly, the DIME-Asp in human MCU was mutated to Ala (D261A), and the mutant was expressed in MCU-KO HEK 293 cells for analysis. Surprisingly, a standard mitochondrial $Ca^{2+}$ uptake assay shows that D261A MCU is capable of importing $Ca^{2+}$ (10 µM), with the rate of transport unaffected by adding 100 mM $Na^+$, which has an ionic radius virtually identical to $Ca^{2+}$ (*Figure 3* and *Figure 3—figure supplement 1*). A quantitative $^{45}Ca^{2+}$ flux experiment (*Tsai et al., 2016*) performed in 10 µM $Ca^{2+}$ shows that D261A slows MCU's $Ca^{2+}$ transport by only 3.8-fold (*Figure 3—figure supplement 2*), an effect remarkably small considering the critical position of this residue in the pore (*Baradaran et al., 2018*; *Fan et al., 2018*; *Nguyen et al., 2018*; *Yoo et al., 2018*). In contrast, mutating the DIME-Glu (E264) to Ala, Asn, or Gln abolishes uniporter function (*Figure 3* and *Figure 3—figure supplements 2* and *3*), as expected from its key role in coordinating $Ca^{2+}$ in the selectivity filter (*Baradaran et al., 2018*; *Fan et al., 2018*; *Nguyen et al., 2018*; *Yoo et al., 2018*). To further pursue these observations, D261 was mutated to all other 18 amino-acids. Only D, E and A at this position support $Ca^{2+}$ transport (*Figure 3—figure supplement 3*).

We then performed CoIP to test how wild-type (WT) MICU1 binding responds to MCU mutations at D261 and E264. Results show that MICU1 binds WT, D261E, and E264A MCU, but not D261A or D261Q (*Figure 4*). Although the D261Q mutant cannot transport $Ca^{2+}$, it still assembles as oligomers (*Figure 4—figure supplement 1*), suggesting that the mutation does not compromise MCU's structural integrity. These results demonstrate that the DIME-Asp mediates MCU interaction with MICU1, instead of contributing essentially to $Ca^{2+}$ permeation.

It was observed that D261A loses sensitivity to a potent and specific uniporter inhibitor Ru360 (*Matlib et al., 1998*),20 (*Figure 3B*). This is consistent with the thought that D261 contributes to a

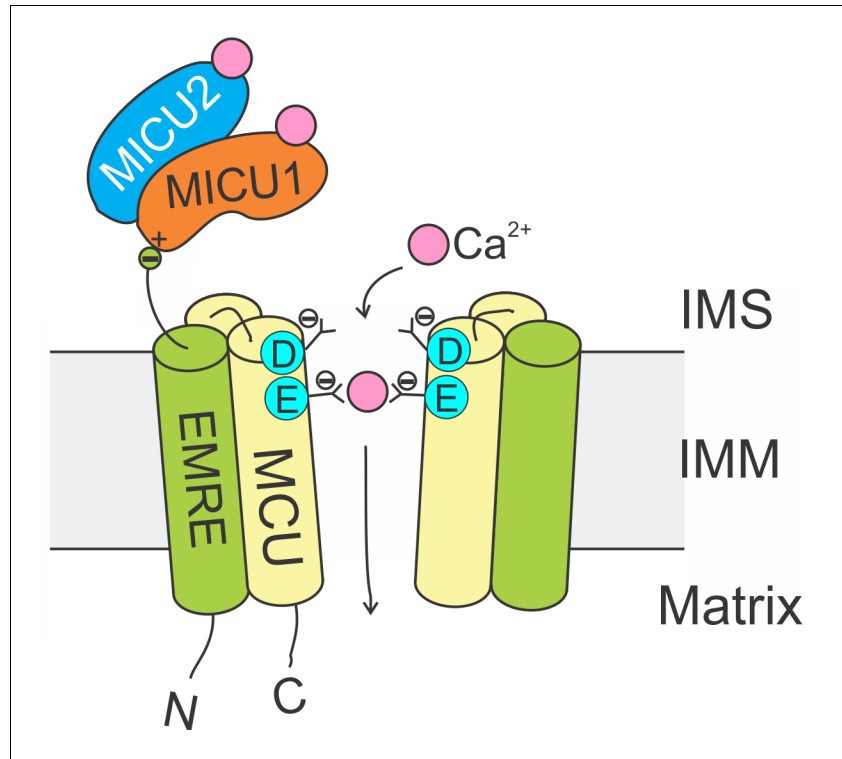

**Figure 1.** Molecular assembly of the mitochondrial $Ca^{2+}$ uniporter. The MCU protein assembles into a tetrameric $Ca^{2+}$ pathway across the inner mitochondrial membrane (only two subunits are illustrated to reveal the $Ca^{2+}$ pore). Conserved Asp and Glu residues in MCU's DIME signature sequence form two parallel side-chain carboxylate rings at the IMS entrance of the pore to coordinate $Ca^{2+}$. The EMRE protein binds to MCU and MICU1 via its TM helix and C-terminal tail, respectively. When an intracellular $Ca^{2+}$ signal arrives at the IMS surface of the uniporter, $Ca^{2+}$ binding to MICUs leads to activation of the uniporter to transport $Ca^{2+}$ into the matrix.
DOI: https://doi.org/10.7554/eLife.41112.002

Ru360 site in MCU (*Arduino et al., 2017*; *Cao et al., 2017*), and implies that MICU1 and Ru360 inhibitory sites overlap. A previous study shows that the S259A mutation diminishes Ru360 inhibition (*Baughman et al., 2011*), raising a possibility that S259 might also be involved in MICU1 binding. We confirm that S259A reduces Ru360 inhibition of the uniporter by 82 ± 3% (*Figure 4—figure supplement 2*), and show that this mutation indeed destabilizes the MCU-MICU1 complex (*Figure 4—figure supplement 2*), albeit to a lesser degree than D261A. It thus appears that MCU and MICU1 form a multi-residue contact surface containing S259 and D261 in MCU, with the latter playing a more critical role in mediating tight MCU-MICU1 interactions.

## Electrostatic interactions between MCU and MICU1

As DIME-Asp appears as a fourfold ring of negative charges facing the IMS, it is tempting to picture MICU1 as a classic pore-blocker (*Banerjee et al., 2013*; *Park and Miller, 1992*) electrostatically stabilized on MCU's ion entryway. This picture is strongly supported by the observation that the MCU-MICU1 interaction can be weakened or strengthened by raising or lowering ionic strength, respectively (*Figure 5A*). In contrast, neither dissociation of the MICU1-MICU2 dimer nor the 1D4-tag and anti-1D4 antibody epitope interaction is affected by varying ionic strength (*Figure 5—figure supplement 1*). To search MICU1 for electrostatic binding partners of the DIME-Asp, we launched an Ala mutagenesis screen targeting 18 conserved Arg or Lys residues in human MICU1 (*Figure 5—figure supplement 2*). Only R119 and R154, two residues closely spaced on the protein's surface (*Wang et al., 2014*), were found to abolish MCU binding upon mutation to Ala (*Figure 5B* and *Figure 5—figure supplement 3*). These mutants, like WT MICU1, form heterodimers with MICU2 (*Patron et al., 2014*) (*Figure 5—figure supplement 4*), indicating proper protein folding. Moreover,

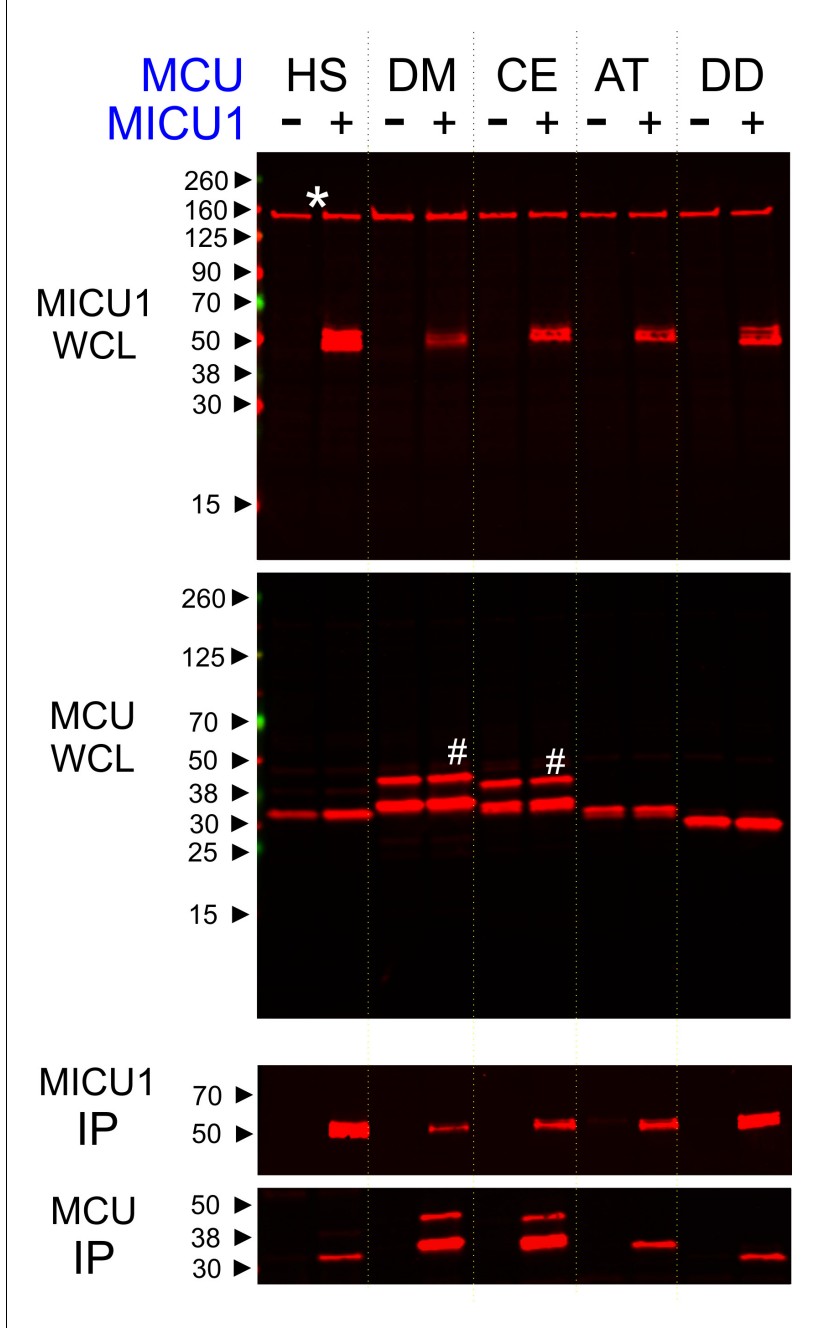

**Figure 2.** Conserved MCU-MICU1 interactions. 1D4-tagged MCU homologues from various species (HS: *Homo sapiens*, DM: *Drosophila Melanogaster*, CE: *Caenorhabditis elegans*, AT: *Arabidopsis thaliana*, and DD: *Dictyostelium discoideum*) were expressed in the presence or absence of FLAG-tagged WT human MICU1 in MCU/EMRE-KO cells. MICU1 was immobilized in FLAG-affinity resins to pull down MCU. Anti-FLAG and anti-1D4 antibodies were used to detect MICU1 and MCU, respectively. SDS-PAGE was performed under reducing conditions. *WCL*: whole cell lysate. *IP*: immunoprecipitation. *Asterisk*: non-specific Western signals. *Hash*: MCU homologues that contain untruncated mitochondrial-targeting sequences.
DOI: https://doi.org/10.7554/eLife.41112.003

R119K or R154K mutants remain associated with MCU, while Glu or Gln substitutions in these two positions strongly disrupt MCU binding (*Figure 5—figure supplement 5*). Neither of the two Arg residues is present in MICU2, and MICU2 is indeed unable to complex with MCU (*Figure 5C*). Taken

together, the data suggest that R119 and R154 in MICU1 mediate electrostatic interactions with the DIME-Asp in MCU.

## Functional roles of the MCU-MICU1 interaction

We have thus far utilized transiently expressed WT or mutant MICU1 to identify molecular determinants of the MCU-MICU1 interaction. However, as MICU1 exclusively forms a disulfide-connected heterodimer with MICU2 in mammalian cells (*Patron et al., 2014*; *Petrungaro et al., 2015*), it is necessary to exclude the possibility that dimerization with MICU2 could fundamentally alter how MICU1 contacts MCU. Accordingly, we employed MCU to pull down native MICUs. Results show that the D261A mutation disrupts MCU association with the physiological MICU1-2 heterodimer (*Figure 6*), indicating that the MICU2-bound form of MICU1 still interacts with MCU via the DIME-Asp.

As binding of MICU1 to the DIME-Asp would likely block the uniporter's pore, we hypothesize that MICU1 shuts the uniporter in resting $Ca^{2+}$ ($<1\ \mu M$) through this particular interaction. This hypothesis predicts that (1) raising $Ca^{2+}$ to micromolar levels would disrupt MCU's association with the MICU1-2 heterodimer, and that (2) perturbing the MCU-MICU1 interaction by mutating the DIME-Asp or R119/R154 would prevent MICU1 from shutting the uniporter. Indeed, CoIP experiments show that supplying $10\ \mu M\ Ca^{2+}$ breaks the MCU-MICU1-MICU2 complex (*Figure 6*). The $^{45}Ca^{2+}$ flux assay described above was subsequently used to quantify mitochondrial uptake under a low $Ca^{2+}$ (0.5 $\mu M$) condition. In WT cells, little $Ca^{2+}$ entry ($1.6 \pm 0.9$ pmol/min/$10^6$ cells) into mitochondria was detected (*Figure 7A*). As expected, MICU1-KO induces robust $Ca^{2+}$ influx ($205 \pm 11$

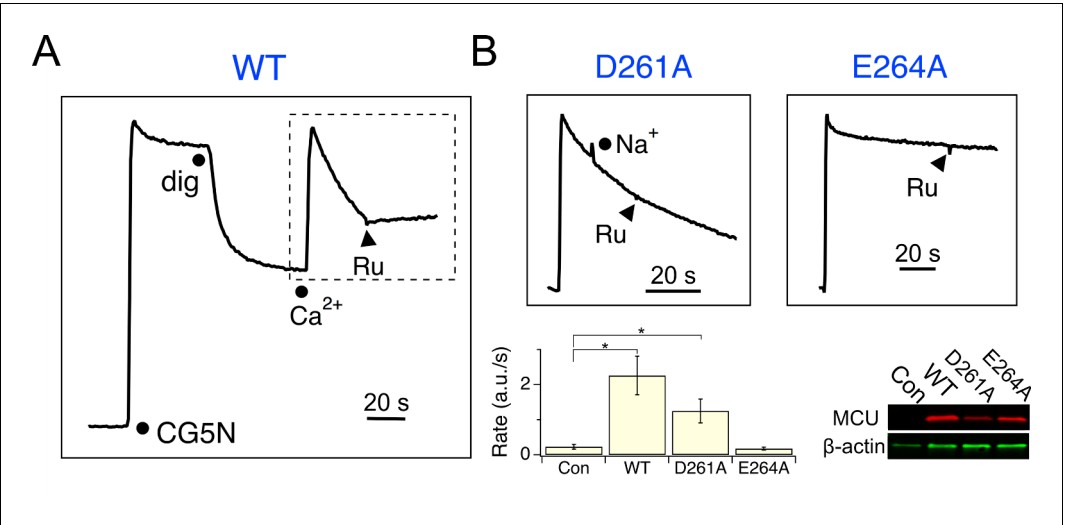

**Figure 3.** Functional analysis of MCU. (**A**) A fluorescence-based mitochondrial $Ca^{2+}$ uptake assay. MCU-KO HEK293 cells, transiently expressing WT MCU, were permeabilized with digitonin (dig) in the presence of an extracellular $Ca^{2+}$ indicator Calcium Green-5N (CG5N). Adding $10\ \mu M\ CaCl_2$ leads to an immediate increase of fluorescence, followed by a signal decline reflecting uniporter-mediated $Ca^{2+}$ uptake. Ru360 (Ru) was added to inhibit the channel. In subsequent experiments, only traces obtained after applying $Ca^{2+}$ (dashed box) are presented. (**B**) The activity of D261A or E264A mutants. These mutants were expressed in MCU-KO cells, with 100 mM NaCl added during $Ca^{2+}$ uptake to test if the channel can select $Ca^{2+}$ against $Na^+$. The bar chart summarizes the initial rate of $Ca^{2+}$ uptake, and the western blot compares expression levels of MCU constructs. *Con:* untransfected cells. *p<0.01.

DOI: https://doi.org/10.7554/eLife.41112.004

The following figure supplements are available for figure 3:

**Figure supplement 1.** The response of WT MCU to $Na^+$.
DOI: https://doi.org/10.7554/eLife.41112.005

**Figure supplement 2.** Quantification of uniporter $Ca^{2+}$ transport.
DOI: https://doi.org/10.7554/eLife.41112.006

**Figure supplement 3.** The activity of D261 or E264 MCU mutants.
DOI: https://doi.org/10.7554/eLife.41112.007

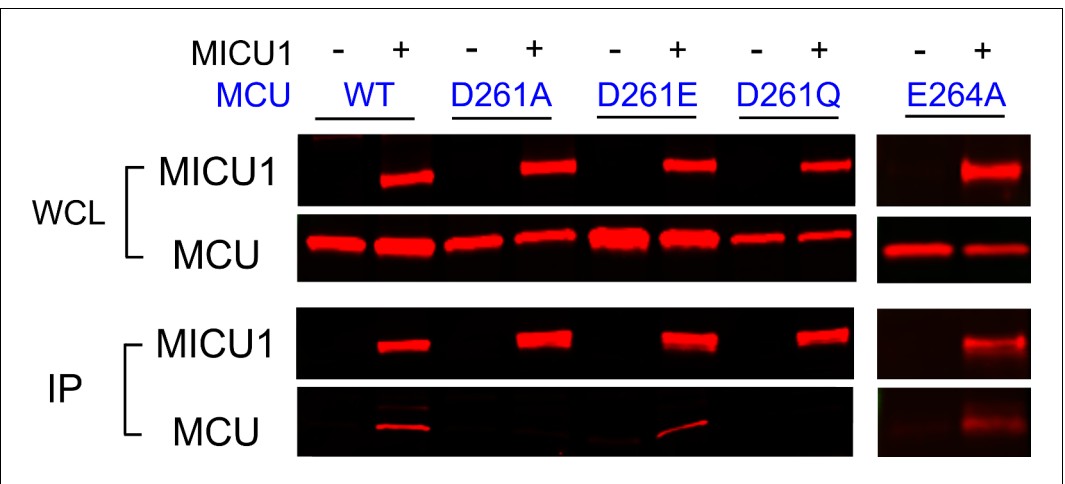

**Figure 4.** The impact of D261 or E264 mutations on MICU1 binding. FLAG-tagged WT MICU1 was used to pull down various MCU mutants co-expressed in MCU/EMRE-KO cells.

DOI: https://doi.org/10.7554/eLife.41112.008

The following figure supplements are available for figure 4:

**Figure supplement 1.** Oligomerization of D261 mutants.

DOI: https://doi.org/10.7554/eLife.41112.009

**Figure supplement 2.** The role of S259 in Ru360 inhibition and MICU1 binding.

DOI: https://doi.org/10.7554/eLife.41112.010

pmol/min/$10^6$ cells), a phenotype partially reversed by expressing WT MICU1 ($53 \pm 4$ pmol/min/$10^6$ cells, *Figure 7A*). We then introduced WT or D261A MCU into MCU-KO cells. In low $Ca^{2+}$, WT MCU exhibits no activity ($1.7 \pm 0.5$ pmol/min/$10^6$ cells) while D261A mediates a $Ca^{2+}$ influx ($34 \pm 5$ pmol/min/$10^6$ cells) 6.2-fold slower than that observed in MICU1-KO cells (*Figure 7B*). A few factors might underlie the rather small magnitude of the D261A-mediated $Ca^{2+}$ uptake: (1) this mutant is 3.8-fold slower than WT MCU (*Figure 3—figure supplement 2*), (2) our transfection efficiency is ~80%, and (3) other residues (e.g., S259) are also involved in MICU1 binding. A S259A/D261A double mutant was constructed to further disrupt the MCU-MICU1 interface, but unfortunately its function could not be analyzed due to a low expression level (*Figure 4—figure supplement 2*). The finding that D261A catalyzes unregulated $Ca^{2+}$ flux in submicromolar $Ca^{2+}$ argues strongly that MICU1 must contact MCU to gate the uniporter. Lastly, we tested R119 or R154 mutants in MICU1-KO cells. All of these, except for R154Q, are less competent than WT MICU1 in restoring $Ca^{2+}$ regulation of the uniporter (*Figure 7C*), a result confirming the critical role of the MCU-MICU1 interaction in $Ca^{2+}$-activation of the uniporter.

## Discussion

The mitochondrial $Ca^{2+}$ uniporter plays a crucial physiological role of regulating cytoplasmic $Ca^{2+}$ signals and controlling mitochondrial metabolic and apoptotic pathways. These processes require the uniporter to remain strictly quiescent in resting cellular conditions. Here, we propose a mechanism (*Figure 8*) in which MICU1 shuts the uniporter by binding to the DIME-Asp side-chain carboxylate ring to block the IMS entrance of the MCU pore. Upon arrival of intracellular $Ca^{2+}$ signals, $Ca^{2+}$ binding to MICU1 at its EF hands disrupts this interaction, thus leading to opening of this $Ca^{2+}$-activated $Ca^{2+}$ channel.

The EMRE subunit, which binds both MCU and MICU1 (*Tsai et al., 2016*), plays an important role in this mechanism. It has been shown that the EMRE-MICU1 interaction is necessary to prevent MICU1 dissociation from the uniporter complex (*Tsai et al., 2016*). We can now understand this observation in light of new results here: When MCU and MICU1 separate due to $Ca^{2+}$ elevation, EMRE's tether to MICU1 would prevent this subunit from dissociating away. Thus, once the $Ca^{2+}$ signal is over, MICU1 could rapidly bind to MCU to terminate $Ca^{2+}$ influx (*Figure 8*).

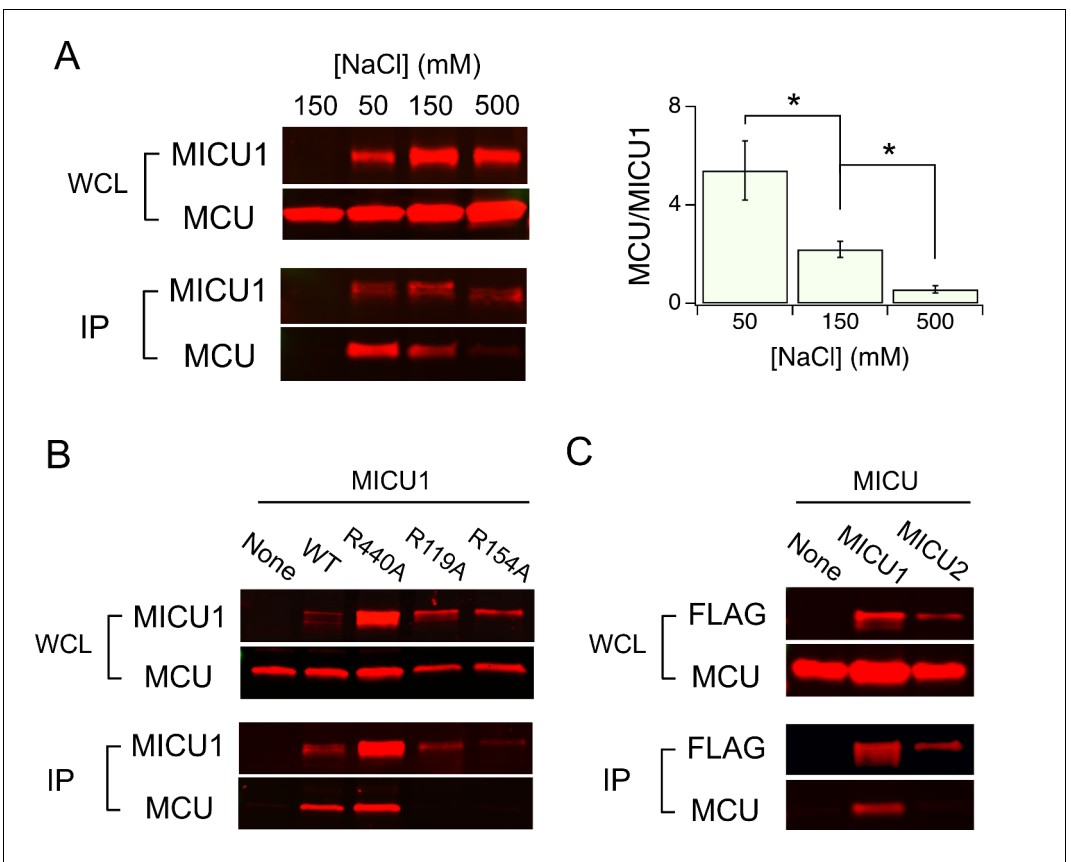

**Figure 5.** Electrostatic interactions between MCU and MICU1. (**A**) Modulation of MCU-MICU1 complex stability by ionic strength. WT MCU and MICU1 were expressed in MCU/EMRE-KO cells, and CoIP experiments were performed in the presence of 50, 150, or 500 mM of NaCl. The IP signal of MCU was normalized to that of MICU1, with the ratio presented in the bar chart. (**B**) The effect of MICU1 Arg mutations on MCU binding. (**C**) A CoIP experiment testing if MCU and MICU2 form complexes. MICU2 was FLAG-tagged to precipitate WT MCU in MCU/EMRE-KO cells. *$p < 0.05$.

DOI: https://doi.org/10.7554/eLife.41112.011

The following figure supplements are available for figure 5:

**Figure supplement 1.** The effect of varying ionic strength on protein-protein interactions.
DOI: https://doi.org/10.7554/eLife.41112.012
**Figure supplement 2.** Multiple sequence alignment of MICU1.
DOI: https://doi.org/10.7554/eLife.41112.013
**Figure supplement 3.** MICU1 mutagenesis screen.
DOI: https://doi.org/10.7554/eLife.41112.014
**Figure supplement 4.** MICU1-MICU2 Interactions.
DOI: https://doi.org/10.7554/eLife.41112.015
**Figure supplement 5.** The effect of R119/R154 mutations on MCU-MICU1 complex formation.
DOI: https://doi.org/10.7554/eLife.41112.016

In this model, MICU2 does not directly contact MCU to block the channel (*Figure 8*). This is consistent with previous work (*Payne et al., 2017*) (but *c.f.* other references, *Plovanich et al., 2013*; *Kamer et al., 2017*) showing that MICU2 is not required to gate the uniporter closed. A fundamental issue for the future would be to determine the function of MICU2 (*Payne et al., 2017*). MICU2 likely plays non-redundant roles, as MICUs are present exclusively in the form of MICU1-2 heterodimers in mammalian cells (*Patron et al., 2014*; *Petrungaro et al., 2015*), and as MICU2 depletion induces severe neuronal and cardiac pathologies (*Bick et al., 2017*; *Shamseldin et al., 2017*).

During the revision of this manuscript, Paillard *et al.* published an article (*Paillard et al., 2018*) showing that depletion of MICU1 sensitizes the uniporter to Ru360 inhibition. The interpretation was

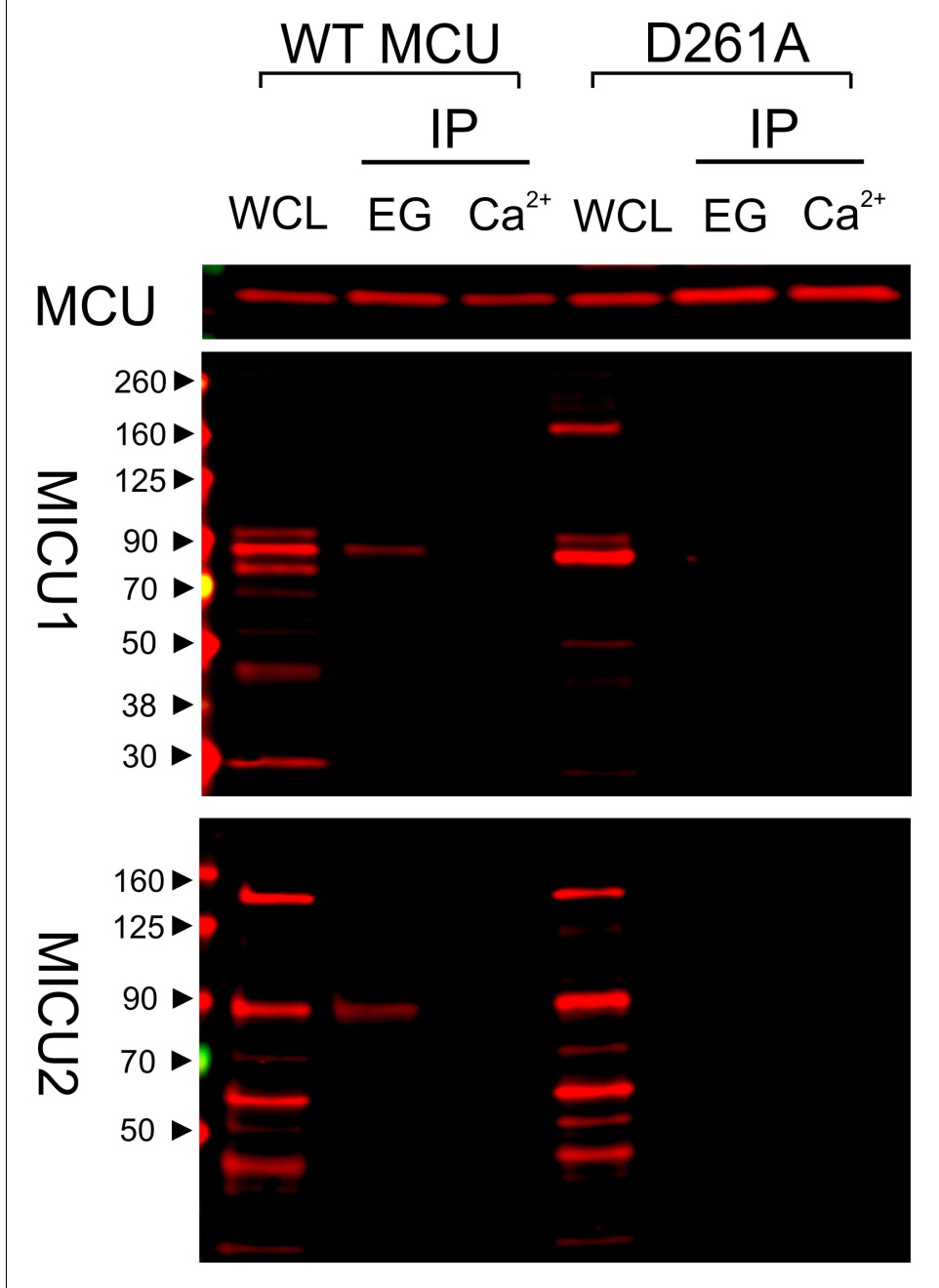

**Figure 6.** Ca$^{2+}$-dependent interaction between MCU and the MICU1-2 heterodimer. 1D4-tagged WT or D261A MCU was expressed in WT HEK cells. The cell lysate, after a portion was taken for whole-cell lysate (WCL) analysis, was split into two for CoIP under Ca$^{2+}$-free (EG, 1 mM EGTA) or 10 μM Ca$^{2+}$ conditions. MCU was used to pull down the native, disulfide-connected MICU1-2 heterodimer (*Patron et al., 2014*; *Petrungaro et al., 2015*), which has a molecular weight of ~90 kDa. SDS-PAGE was performed in non-reducing environments. MICU1 and MICU2 were detected using anti-MICU1 and -MICU2 antibodies, respectively. WCL signals of MICU1 and MICU2 are not as clean as in previous images (*e.g.*, *Figure 2*) due to the low abundance of native MICUs and lower qualities of these polyclonal MICU1 and MICU2 antibodies.

DOI: https://doi.org/10.7554/eLife.41112.017

that MICU1 competes for the Ru360 inhibitory site, known to be formed by the DIME-Asp (*Arduino et al., 2017*; *Cao et al., 2017*). It follows that MICU1 must control the uniporter by interacting with the Asp ring. Our results similarly indicate that MICU1 and Ru360 sites in MCU likely

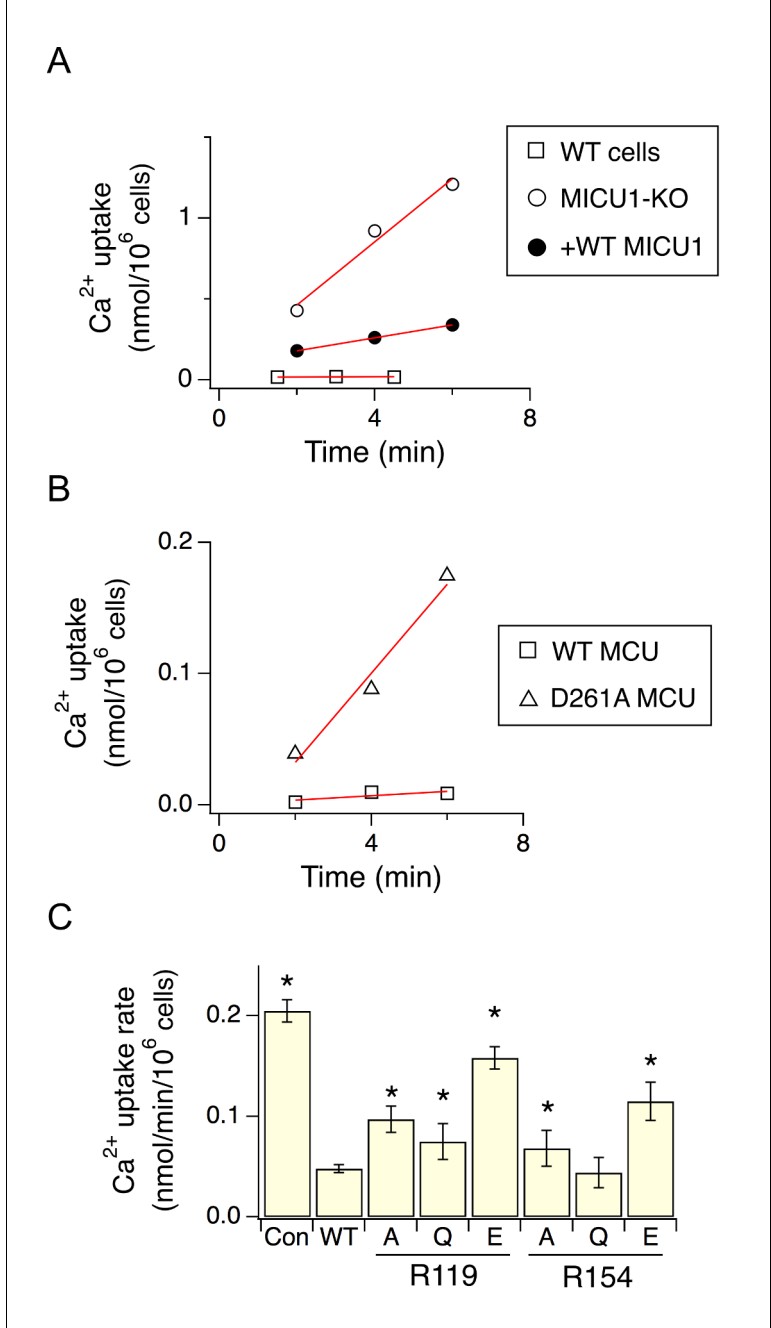

**Figure 7.** The effect of D261 or R119/R154 mutations on the regulatory function of MICU1. (**A**) Mitochondrial $Ca^{2+}$ uptake in a low $Ca^{2+}$ (0.5 μM) condition. Each data point represents a measurement of $^{45}Ca^{2+}$ transported into mitochondria by the uniporter at a specific time point. These data points were fit with a linear function (red lines) to obtain the rate of $Ca^{2+}$ transport. (**B**) The activity of WT or D261A MCU in 0.5 μM $Ca^{2+}$. (**C**) A bar chart summarizing the rate of mitochondrial $Ca^{2+}$ uptake. WT MICU1 or various R119/R154 mutants were expressed in MICU1-KO cells. *Con*: untransfected control. Paired t-test was performed between WT MICU1 and mutants. *p<0.05.

DOI: https://doi.org/10.7554/eLife.41112.018

The following figure supplement is available for figure 7:

**Figure supplement 1.** Data processing in $^{45}Ca^{2+}$ flux experiments.

DOI: https://doi.org/10.7554/eLife.41112.019

overlap, as mutations in DIME-Asp and a nearby Ser (S259 in human MCU) perturb both Ru360 inhibition and MICU1 binding.

Paillard *et al.* further proposed that MICU1 uses a DIME-interacting domain (DID) that contains one Lys and two Args (K438, R440, and R443 in human MICU1) to bind MCU. However, it was also shown that with all these residues mutated to Ala, a portion of the MCU-MICU1 complex (~30% of that observed using WT MICU1) remains associated after tens of minutes of incubation in CoIP experiments. This result, which agrees with our finding that R440A or R443A MICU1 forms stable complexes with MCU (*Figure 5—figure supplement 3*), raises a possibility that the DID sequence might not play direct roles in mediating tight MCU-MICU1 interactions.

Our model instead posits that MICU1 uses two closely-spaced Arg in the N-terminal domain (*Wang et al., 2014*) (R119 and R154 in human MICU1) to bind the DIME-Asp (*Figure 8*). This picture is supported by the observations that, consistent with electrostatic interactions, the stability of the MCU-MICU1 complex can be modulated by varying the ionic strength, and that MICU2, which lacks these Args, is unable to bind MCU (the DID sequence is present in both MICU1 and MICU2). The crucial roles of these two Args in MCU binding are further highlighted by the fact that they are the only two basic residues that are conserved in MICU1 homologues in animals, plants, and protists (*Figure 5—figure supplement 2*), in which MCU and MICU1 co-evolve (*Bick et al., 2012*). However, we hasten to point out that, despite these observations, future biochemical and structural work is still required to determine the detailed chemistry that governs MICU1 interactions with the DIME-Asp.

It is known that the uniporter uses a classical multi-ion pore mechanism (*Kirichok et al., 2004*; *Almers et al., 1984*; *Hess and Tsien, 1984*) to select $Ca^{2+}$ against >1000-fold more abundant cations such as $Na^+$. In this mechanism, $Ca^{2+}$ binding to a high-affinity site blocks permeation of other cations, while entry of a second $Ca^{2+}$ knocks off the bound $Ca^{2+}$ through electrostatic repulsion to enable high $Ca^{2+}$ flux. New structures of MCU led to the hypothesis that the DIME-Glu forms the high-affinity site (S2) to coordinate a dehydrated $Ca^{2+}$, while the DIME-Asp forms a second, low-affinity $Ca^{2+}$ site (S1) (*Baradaran et al., 2018*; *Fan et al., 2018*; *Nguyen et al., 2018*; *Yoo et al., 2018*). We systematically mutated DIME-Asp (D261) in human MCU and found that most mutations abolish channel function, an outcome not unexpected considering the critical position of D261 in the pore. The fact that D261A exhibits a comparable activity as WT, however, raises a possibility that other $Ca^{2+}$ sites might be present in proximity to S2 to mediate the electrostatic repulsion required for high $Ca^{2+}$ throughput of the uniporter.

In conclusion, the current study provides a working model to understand how intracellular $Ca^{2+}$ signals control the activity of the uniporter in the molecular level. Major challenges still lie ahead,

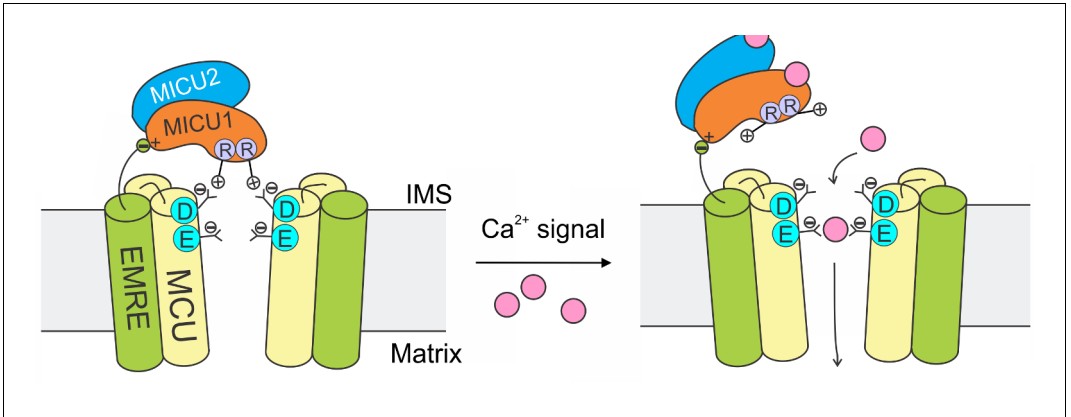

**Figure 8.** A model of $Ca^{2+}$-dependent gating of the uniporter. In resting cellular conditions, MICU1 shuts the uniporter by inserting Arg fingers into MCU's Asp ring to occlude the pore. $Ca^{2+}$ activates the channel by binding to MICUs to disrupt this MCU-MICU1 interaction. MICU2 forms a heterodimer with MICU1, but does not directly contact MCU. EMRE plays dual functional roles: it binds to MCU to enable $Ca^{2+}$ permeation, and also interacts with MICU1 to maintain tight association of the MICU1-2 heterodimer with the uniporter during $Ca^{2+}$ stimulation.
DOI: https://doi.org/10.7554/eLife.41112.020

including to understand MICU2's physiological role, to determine how $Ca^{2+}$ disrupts the MCU-MICU1 interaction, and to examine the individual roles of MICU1's two EF hands in channel activation. New electrophysiological and structural tools (*Baradaran et al., 2018*; *Fan et al., 2018*; *Nguyen et al., 2018*; *Yoo et al., 2018*; *Tsai and Tsai, 2018*) will open exciting opportunities to address these in the future.

# Materials and methods

## Key resources table

| Reagent type or resource | Designation | Source or reference | Identifiers | Additional information |
|---|---|---|---|---|
| Cell line | HEK 293T | ATCC | Cat # CRL-3216 | |
| Cell line | MCU-KO HEK 293T | PMID:27099988 | | |
| Cell line | MCU/EMRE-KO HEK 293T | PMID:27099988 | | |
| Cell line | MICU1-KO | PMID:28396416 | | |
| Primary Antibody | Mouse anti-FLAG | Sigma-Aldrich | Cat # F1804 | Western 1:10000 |
| Primary Antibody | Mouse anti-V5 | ThermoFisher | Cat # R960-25 | Western 1:5000 |
| Primary Antibody | Mouse anti-β actin | Santa Cruz | Cat # 69879 | Western 1:500 |
| Primary Antibody | Rabbit anti-MICU1 | Sigma-Aldrich | Cat # HPA037480 | Western 1:5000 |
| Primary Antibody | Rabbit anti-EFHA1 (MICU2) | Abcam | Cat # ab101465 | Western 1:10000 |
| Primary Antibody | Mouse anti-1D4 | PMID:6529569 | | Western 50 ng/mL |
| Primary Antibody | Mouse anti-C8 | PMID:8068416 | | Western 50 ng/mL |
| Secondary Antibody | IRDye 680RD goat anti-rabbit IgG | Li-Cor | Cat # 925–68073 | Western 1:10000 |
| Secondary Antibody | IRDye 680RD goat anti-mouse IgG | Li-Cor | Cat # 925–68072 | Western 1:15000 |
| Chemical compound | Ru360 | PMID:2036363 | | |
| Chemical compound | $^{45}CaCl_2$ | PerkinElmer | Cat # NEX01300 | |
| Commercial kit | Lipofectamine 3000 | ThermoFisher | Cat # L3000015 | |
| Commercial kit | Anti-FLAG M2 affinity gel | Sigma-Aldrich | Cat # A2220 | |
| Commercial kit | CNBr-activated Sepharose 4B | GE Healthcare | Cat # 17043001 | |
| Software | Igor Pro 7 | WaveMetrics | | Figure production and data fitting |
| Software | ImageStudio 5 | Li-Cor | | Western-blot quantification |
| Software | Clustal Omega | PMID:21988835 | | Sequence alignment |
| Software | Excel (office 365) | Microsoft | | t-test |

## Reagents, cell culture, and molecular biology

Reagents were purchased at the highest grade available. Ru360 was synthesized in-house following a previously published protocol (*Ying et al., 1991*). Genes encoding uniporter subunits were cloned into a pcDNA 3.1 (+) expression vector. Site-directed mutagenesis was performed using a Quick-Change kit (Agilent) and confirmed with sequencing. All MCU constructs used here contain a C-terminal 1D4 tag (TETSQVAPA) for Western detection. Similarly, MICU1 is tagged with a C-terminal FLAG (DYKDDDDK), and MICU2 with a C-terminal FLAG or V5 (GKPIPNPLLGLDST). Sequences of these have been reported in a previous manuscript (*Tsai et al., 2016*).

HEK 293 cells, obtained from ATCC and authenticated by short tandem repeat profiling, were cultured in Dulbecco's modified Eagle's medium (Gibco) supplemented with 10% FBS, and were incubated at 37°C with 5% $CO_2$. Mycoplasma infection was routinely ruled out using an ATCC PCR detection kit (30–1012K). CRISPR knockout cell lines have been established in our previous work (*Tsai et al., 2016*; *Tsai et al., 2017*). Transient transfection was performed using Lipofectamine 3000 (ThermoFisher), following the manufacturer's instructions. Cells were harvested for experiments 24–30 hr after transfection.

## Co-immunoprecipitation (CoIP)

All CoIP experiments were performed at 4°C. Transfected cells in 2 wells of a 6-well plate were lysed in 0.5 mL solubilization buffer (SB, 100 mM NaCl, 20 mM Tris, 1 mM EGTA, 5 mM DDM, pH 7.5-HCl) supplemented with an EDTA-free protease inhibitor cocktail (cOmplete Ultra, Roche). The lysate was clarified by spinning down. 50 µL of the supernatant was removed, with total protein concentration determined using a BCA assay (Thermo-Fisher) and 10 µg of protein used for whole-cell lysate (WCL) analysis. Then, 25 µL of FLAG (Sigma-Aldrich, A2220)- or 1D4-conjugated beads (50% slurry) were added to the rest of the supernatant for a 30 min batch binding process. The beads were then collected on a spin column, washed with 2 mL of SB, and then eluted with 0.15 mL SDS loading buffer. 10–20 µL of the elute was used for SDS-PAGE, with 5% of 2-mercaptoethanol used to produce reducing conditions. The whole CoIP procedure was completed within 45 min after cell lysis (prolonged incubation of >2 hr could lead to complete dissociation of uniporter subcomplexes). 1D4-affinity gel was produced in house using 25 mg 1D4 antibody per 1 g of CNBr-activated Sepharose 4B resin (GE Healthcare).

To perform Western blot, proteins on SDS gels were transferred to low-fluoresce PVDF membranes (EMD-Millipore), which were then blocked in a TBS-based Odyssey blocking buffer (Li-Cor), and incubated with primary antibodies in TBST (TBS +0.075% Tween-20) at 4°C overnight. Then, after a 1 hr incubation with infrared fluorescent secondary antibodies in TBST at room temperature, signals were acquired using an Odyssey CLx imaging system (Li-Cor), and analyzed with an Image-Studio software (Li-Cor version 5.0). Unless specified, MCU and MICU1 were detected using $\alpha$−1D4 and $\alpha$-FLAG antibodies, respectively. See the key resources table for antibodies and dilutions. 1D4 and C8 antibodies were produced in house.

## Mitochondrial $Ca^{2+}$ flux assays

For the fluorescence-based assay, $2 \times 10^7$ HEK 293 cells were suspended in 10 mL of wash buffer (WB, 120 mM KCl, 25 mM HEPES, 2 mM $KH_2PO_4$, 1 mM $MgCl_2$, 50 µM EGTA, pH 7.2-KOH), pelleted, and then resuspended in 2.5 mL of recording buffer (RB, 120 mM KCl, 25 mM HEPES, 2 mM $KH_2PO_4$, 5 mM succinate, 1 mM $MgCl_2$, 5 µM thapsigargin pH 7.2-KOH). 2 mL of the cell suspension were placed in a stirred quartz cuvette in a Hitachi F-2500 spectrophotometer (ex: 506 nm, ex-slit: 2.5 nm, em: 532 nm, em-slit: 2.5 nm, sampling rate: 2 Hz). Reagents were added into the cell suspension in the following order: 0.5 µM calcium green 5N (Thermo-Fisher C3737), 30 µM digitonin (Sigma-Aldrich D141), 10 µM $CaCl_2$, and 75 nM Ru360. Upon adding $Ca^{2+}$, fluorescent signals would increase by 200 to 300 a.u. Without adding Ru360, the signal would eventually drop to a steady-state level roughly the same as that before $Ca^{2+}$ addition. Quantification of data is done by linear fit to the fluorescent signal between 10 s and 15 s after adding $Ca^{2+}$.

For the $^{45}Ca^{2+}$ based assay, $1.2–2.4 * 10^6$ viable cells were suspended in 1 mL WB, spun down, and then resuspended in 120 µL WB, supplemented with 5 µM thapsigargin (Sigma-Aldrich, T9033) and 30 µM digitonin. To initiate mitochondrial $Ca^{2+}$ uptake, 100 µL cell suspension was transferred to 400 µL low-$Ca^{2+}$ flux buffer (RB +0.69 mM EGTA, 0.5 mM $CaCl_2$, 15 µM $^{45}CaCl_2$, 30 µM digitonin,

5 µM thapsigargin, pH 7.2-KOH) or high-$Ca^{2+}$ flux buffer (RB +20 µM $^{45}CaCl_2$, 30 µM digitonin, 5 µM thapsigargin, pH 7.2-KOH). At desired time points, $Ca^{2+}$ uptake was terminated by adding 100 µL of the sample to 5 mL ice-cold WB, and then filtered through 0.45 µM nitrocellulose membranes (Sigma-Aldrich WHA10402506) on a vacuum filtration manifold (EMD-Millipore model 1225). The membrane was washed immediately with 5 mL ice-cold WB, and later transferred into scintillation vials for counting. Nonspecific signals were measured using samples containing 75 nM Ru360 or using untransfected cells (for the Ru360-insensitive D261A mutant), and were subtracted to yield uniporter-specific $Ca^{2+}$ transport. In a typical experiment, readings of $^{45}Ca^{2+}$ in three time points were fit with a linear function to generate the rate of $Ca^{2+}$ transport (*e.g.*, *Figure 7A*). Rates obtained from at least three independent experiments were then averaged for data presentation (see *Figure 7—figure supplement 1* for examples of the data analysis process). For experiments comparing WT and D261A, 1 µg WT DNA or 2.2 µg D261A DNA was used for transfection to ensure similar expression levels of these two constructs. Moreover, cells were harvested within 24 hr after transfection to avoid a molecular excess of overexpressed MCU over native MICU1. $^{45}Ca^{2+}$ radioisotope was obtained from PerkinElmer, and has a specific activity of 12–15 mCi/mg.

## Sequence analysis and statistics

Sequences of MICU1 homologues were collected using PSI-BLAST. Multiple sequence alignment was performed using the Clustal Omega online server (*Sievers et al., 2011*).

All experiments were repeated in at least three independent experiments, and the data were presented as mean ±standard error of the mean (SEM). Statistical analysis was performed using Student's t-test, with significance defined as $p < 0.05$.

## Acknowledgements

We thank Carole Williams for technical assistance in molecular biology, and Dr. Christopher Miller for critical reading of this manuscript as well as providing unconditional support during the development of this project. We thank Dr. Vamsi Mootha for kindly providing an independent strain of MICU1-KO cells for us to verify results in *Figure 7*. CWT and MFT are partly supported by the NIH grant R01-GM129345. The authors declare no conflict of interests.

## Additional information

### Funding

| Funder | Grant reference number | Author |
|---|---|---|
| National Institute of General Medical Sciences | R01-GM129345 | Chen-Wei Tsai Ming-Feng Tsai |

The funders had no role in study design, data collection and interpretation, or the decision to submit the work for publication. The funders pay for the authors' salary and other research expenses.

### Author contributions

Charles B Phillips, Chen-Wei Tsai, Conceptualization, Data curation, Formal analysis, Validation, Investigation, Writing—review and editing; Ming-Feng Tsai, Conceptualization, Data curation, Formal analysis, Supervision, Funding acquisition, Validation, Investigation, Writing—original draft, Project administration, Writing—review and editing

### Author ORCIDs

Ming-Feng Tsai (iD) https://orcid.org/0000-0003-4277-1885

### Decision letter and Author response

Decision letter https://doi.org/10.7554/eLife.41112.025
Author response https://doi.org/10.7554/eLife.41112.026

## Additional files

### Supplementary files

• Transparent reporting form

DOI: https://doi.org/10.7554/eLife.41112.021

### Data availability

All data generated or analyzed during this study are included in the manuscript and supporting files. Source data for calcium-45 flux experiments are available via Dryad.

The following dataset was generated:

| Author(s) | Year | Dataset title | Dataset URL | Database and Identifier |
|-----------|------|---------------|-------------|-------------------------|
| Phillips C, Tsai C, Tsai M | 2019 | Date from: The conserved aspartate ring of MCU mediates MICU1 binding and regulation in the mitochondrial calcium uniporter complex | https://doi.org/10.5061/dryad.1976kg3 | Dryad Digital Repository, 10.5061/dryad.1976kg3 |

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
