## [Decision Letter]

[Editors’ note: the authors were asked to provide a plan for revisions before the editors issued a final decision. What follows is the editors’ letter requesting such plan.]

Thank you for sending your article entitled "MICU1 interacts with the conserved aspartate ring of MCU to mediate Ca^2+^ activation of the mitochondrial Ca^2+^ uniporter" for peer review at *eLife*. Your article is being evaluated by three peer reviewers, one of whom is a member of our Board of Reviewing Editors, and the evaluation has been overseen by John Kuriyan as the Senior Editor.

Given the list of essential revisions, including new experiments, the editors and reviewers invite you to respond within the next two weeks with an action plan and timetable for the completion of the additional work. We plan to share your responses with the reviewers and then issue a binding recommendation.

Regarding the flux experiments you are correct. These flux experiments have been established earlier in the *eLife* paper of Tsai and Miller you cited above. However, the difference between the *eLife* paper you cited and the current submitted manuscript is that in the previous paper, the interpretations drawn from the assays are within the limits of the assay resolution. As you know, the fluorescent flux assay in question is very qualitative and basically reports whether there is or there is not import of Ca into mitochondria via the uniporter. If a mutation is made in any of the protein components required for function and Ca is no longer imported, the MCU is broken somehow, and the conclusion is drawn that the residue is important for either function or for interaction between the protein components. For example, in the previous paper, the fluorescent assay was used to determine which domains of EMRE are interacting with MCU to support Ca entry into mitochondria by expressing different deletion mutants of EMRE with MCU and concluding that the deletion of the domain that killed the transport was important in the interaction. Similarly, they used the assays to determine which side of the helix interacts with MCU by doing a Trp scanning mutagenesis and finding that the side where all mutations killed the transport, likely interacts with the MCU. These conclusions are, in my opinion, valid and within the resolution of the assay.

In contrast, in the current manuscript, as I previously stated, the authors draw very specific mechanistic interpretations based on this yes/no assay. For example:

1) Mutation of the D in the DIME signature sequence region of MCU to an A did not abolish Ca import but lead to loss of MICU1 interaction (as measured with co-IP). Furthermore, addition of 100 mM Na during the assay did not change the rate of Ca-import. The authors conclude that the D does not contribute to channel selectivity but mediates MICU1 interaction. The problem are the following: first, the assay does not address selectivity at all, but rather whether the mutation kills Ca transport. The authors should consider dropping any mention of the role of D in selectivity. Second, there should be a control experiment showing the effect of Na addition on WT MCU. Third, the finding that mutation of the D to any other amino acid (except E and A) abolished Ca-import is intriguing, and argues that the story is much more complex. Fourth, the authors should also present the coIPs with the D261E MCU mutant, which, according to the authors' hypothesis, is functional and expected to support the interaction with MICU1.

2) The authors claim that the MCU-MICU1 interaction is electrostatic as evidenced by weaker interactions as measured by lower ratios of MCU/MICU1 gel bands post coIPs in higher ionic strength buffer conditions. If the authors want to make this electrostatic claim, they should show both a positive and a negative control experiment. Otherwise, a direct binding interaction assay would be more appropriate in this case.

3) The authors screen for interacting partners for the Asp on MCU by mutating 18 conserved Arg and Lys residues on MICU1 to Ala and checking for interaction using coIPs. They find two Arg that disrupt MCU association and claim that these Rs on MICU1 mediate electrostatic interactions with the D on MCU. Perhaps these claims should be reduced to: the two Rs are important in mediating the interaction between MCU and MICU1, since there is no indication that the Rs interact with the D specifically. Furthermore, if the screen were not limited to positively charged residues, perhaps other residues would pop up as important to this interaction.

4) The authors conclude that the MICU1 interaction with MCU must work by MICU1 directly blocking the MCU pore, predicated on their conclusions that the Rs on MICU1 electrostatically interact with the Ds in MCU, which are located in the pore. Since there is no solid evidence for the Rs interacting with the D, the pore blocking interpretation is similarly weak and I suggest that it is removed.

5) In Figure 6-7, the authors test whether MICU1 interaction with the pore in resting Ca concentrations is necessary to keep the pore closed. They find by coIP that the MICU1-MCU interaction is disrupted in the presence of 10 μm Ca and that in MICU1 knockout cells, Ca fluxes inside mitochondria, while if MICU1 is added back, less Ca is uptaken. If the MICU1 R mutants are added back instead, except for one of them, they are less competent than WT in restoring Ca flux. It would have been useful for the authors to also show that, similar to the R mutants on MICU1, the MCU D to A mutant (the interaction partner of the Rs) is also less competent than WT to restore Ca flux.

In addition, this entire study is predicated on the finding that MICU1 and MCU specifically interact directly and not through EMRE. This is shown exclusively with co IPs, which are usually only preliminary indicators of specific binding. It would be helpful for the solidity of the argument and for all the experiments that follow, for this interaction to be shown with more direct binding assays.

Overall, for the reasons I outlined above, I believe that the authors overreach in their conclusions and interpretations. A few examples: Introduction last paragraph: "These results led to a molecular mechanism in which MICUs open or close the uniporter in response to intracellular Ca signals by physically blocking or unblocking the MCU pore."; first paragraph of Discussion: "Here, we establish a mechanism in which MICU1 shuts the uniporter by binding to the DIME-Asp side chain carboxylate ring to block the IMS entrance of the MCU pore..… Ca-binding to MICU1 at its EF hands disrupts this interaction, thus leading to opening of this Ca-activated Ca channel". Last two paragraphs of the Discussion: ".… a result that demonstrated unambiguously that S1 is not critical for Ca selectivity…..". "..the current study provides insight into the uniporter's ion selectivity mechanisms…". All these statements need to be rethought.

Reviewer #1:

Phillips et al. describe studies on the mitochondrial calcium uniporter (MCU) aimed at examining the mechanism of Ca selectivity, pore gating, and association with the MICU1 and MICU2 subunits. Using coimmunoprecipitation assays, they show that MICU1 and MCU interact directly even in the absence of the EMRE subunit, which was already known to mediate association of MICU1 to the complex. Next, they test the role of the acidic residues in the DIME signature sequence thought to comprise the selectivity filter of the uniporter. They demonstrate that no mutations of the glutamate are tolerated, consistent with the notion that it is central to forming the Ca selective pore. However, the aspartate is not required for selectivity or transport, but is required for gating the pore. Mutations of the DIME-asp also abrogate binding of MICU1 as shown by CoIP assays. Further, they show that MCU-MICU1 association is highly dependent on ionic strength, implicating electrostatic interactions, then identify 2 basic residues on MICU1 that are critical for interaction. These 2 basic residues are absent in MICU2, which the authors show does not directly bind MCU, but associates in vivo through covalent disulfide linkage to MICU1. This was further demonstrated by pulldown using the endogenous MICU1/MICU2 heterodimers and WT and mutant MCU. In the case of WT MCU, both MICU1 and MICU2 are pulled down, whereas in MCU D261A does not pull down either. Finally, they show that mutations to the basic residues of MICU1 similarly reduce the MCU gating ability as mutations to the DIME-asp, supporting the role in this interaction in gating the pore. Based on their experimental results, the authors propose a gating model for the MCU channel that is clear and illustrates their key findings well.

Overall, the study reveals several new features of MCU gating and selectivity such as the role of the DIME-asp in pore gating rather than selectivity, and the direct association of MICU1 and MCU mediated by electrostatic interactions with the DIME-asp. The claims reported by the authors are all supported by the experimental data. The writing is clear and concise but I have some concerns:

Concerns:

1) The authors should clarify how the rates are calculated when the time courses do not saturate. Is there a way to estimate the steady-state levels? One assumes that the errors are large if the time course is significantly slowed down. See D261E in Figure 3—figure supplement 1.

2) Figure 7A. The authors should consider adding more data points so that they are not using just three data points for regression.

Reviewer #2:

In the study entitled "MICU1 interacts with the conserved aspartate ring of MCU to mediate Ca^2+^ activation of the mitochondrial Ca^2+^ uniporter", Phillips et al. seek to understand the mechanism by which the gate-keeping proteins MICU1 and MICU2 confer their regulatory effect on the pore forming subunit MCU of the mitochondrial calcium uniporter. In the absence of structural information on the entire MCU complex, it remains unclear how these MICU proteins regulate the ion conducting pore subunit. Using a combination of pull-down assays and fluorescent calcium uptake assays, the authors identified two conserved arginines (R119 and R154 of the human ortholog) in MICU1 that interact with the highly conserved aspartate residue of the selectivity filter of MCU. This direct interaction allows MICU1 to inhibit the Ca^2+^ uptake function of MCU at low cellular calcium concentrations.

Overall, this study was well done and it addresses a fundamentally important question about how MCU is regulated. Additionally, the body of literature points to MICU1 as an important regulatory component of the uniporter since loss of MICU1 leads to mitochondrial calcium overload from cells to animal models to humans. Thus, the manuscript being considered is highly relevant and appropriate for publication in *eLife*.

Reviewer #3:

The manuscript by Phillips et al. reports on the role of the DIME signature sequence of the mitochondrial uniporter Ca channel, which is to both determine Ca selectivity (the E) and to mediate binding of the accessory subunit MICU1 in the absence of Ca (the D), and blocking the MCU pore. Thus, in the absence of Ca, the authors hypothesize that MICU1 binds to and sterically blocks the pore of the Ca channel, while in the presence of Ca, Ca binding to the EF hand of MICU1 leads to its dissociation from the channel, unblocking the pore and consequently allowing Ca flux.

Major issue:

The topic is interesting, and understanding the mechanism of functioning of this important channel complex is of high impact especially in the view of the recent structures of this channel. However, the detailed mechanistic model that the authors propose, and which I outlined above, is only lightly supported by the experimental data presented in this paper. The experiments are all indirect. They all are either co-immunoprecipitations or fluorescent assays performed on permeabilized HEK cells transiently transfected with WT/mutants of either the pore-forming subunit (MCU) or the interacting proteins (MICU1/2). These experiments appear well-executed but they fall short of demonstrating essentially any of the authors' claims on the MCU mechanism. The obvious components of the mechanism not demonstrated here are: MICU1 as MCU pore blocker, MICU1 unblocking the pore upon Ca-binding, the location of Ca binding for this mechanism, the involvement of the D in the DIME sequence in the interaction between the two proteins, how does Ca break the Asp-Arg mediated interaction between the two proteins, etc.

Other comments:

The Introduction is too short and lacks presentation of the selectivity of the channel (which is actually brought up in the results), as well as even a brief discussion of the existing structure(s). Despite the precision of the conclusions drawn regarding the mechanism of Ca-induced activation of MCU, the authors do not discuss this mechanism from the perspective of the structures, which is I believe necessary.

The conclusion that the Asp in the DIME sequence does not contribute to the channel's selectivity filter is premature as it is based on only one mutant that for some reason still supports Ca uptake and is not affected by Na addition in these particular assays. However, there is no control showing how Na affects these fluorescent assays in the WT or in any of the other mutants, and furthermore, the assay is still indirect.

In the D261A MCU mutant, what keeps the channel closed in normal conditions, if D is crucial to MICU1 binding and if it's this pore block that keeps the channel closed? Is this mutant constitutively active?

I am assuming that the radioactive Ca flux assay was used instead of the fluorescent one for Figure 6 because of the low Ca involved. However, this must be spelled out in the manuscript, because it looked like the fluorescent assay was quite sensitive. Furthermore, more information is needed for both the fluorescent assay and the radioactive assay (the section in the Materials and methods is not detailed enough). For instance, for the radioactive assays, I don't get a sense of what is plotted in Figure 6. Is this only one experiment, since there are no error bars? What's the signal to noise here? What do the signals look like before and after application of the Ru360? Etc…

To summarize, I believe that although the topic is interesting, timely, and of high impact, and the experiments are well-done, the detailed signature-sequence aspartate-mediated protein-protein interaction mechanism of Ca activation of the MCU is an overreach and only indirectly supported by the experimental data presented.

---

## [Author Response]

[Editors’ notes: the authors’ response after being formally invited to submit a revised submission follows.]

Reviewer #1:Overall, the study reveals several new features of MCU gating and selectivity such as the role of the DIME-asp in pore gating rather than selectivity, and the direct association of MICU1 and MCU mediated by electrostatic interactions with the DIME-asp. The claims reported by the authors are all supported by the experimental data. The writing is clear and concise but I have some concerns:Concerns:1) The authors should clarify how the rates are calculated when the time courses do not saturate. Is there a way to estimate the steady-state levels? One assumes that the errors are large if the time course is significantly slowed down. See D261E in Figure 3—figure supplement 1.

This is a very legitimate concern, and we apologize that this issue has not been made clear in the original manuscript. Basically, after adding 10 µM Ca^2+^ (box in Figure 3A), free Ca^2+^ in the extra-mitochondrial solution would eventually drop back to a steady-state level that is roughly the same as that before Ca^2+^ addition. This is because mitochondria have high Ca^2+^ buffering capacity to sequester added Ca^2+^. A main problem in this assay is that the amplitude of fluorescence-signal increase upon adding 10 µM Ca^2+^ can vary from 200 to 300 a.u., presumably due to variations in extra-mitochondrial Ca^2+^ buffering capacity. Moreover, it is difficult to control protein-expression to exactly the same level for all constructs. Thus, the initial rate, presented as a.u./s, is qualitative in nature, and we strictly used this to address yes/no questions. We have now revised the Materials and methods section to highlight these issues. The s.e.m. for D261E is indeed smaller than WT, but in addition to the time course, there could be other factors, such as the consistency of protein expression, that contribute to errors.

2) Figure 7A. The authors should consider adding more data points so that they are not using just three data points for regression.

In Figure 7, we used a quantitative ^45^Ca^2+^ flux assay to determine the rate of mitochondrial Ca^2+^ uptake. In a typical experiment, we obtain readings of ^45^Ca^2+^ at 3 different time points, and then fit the data with a linear function to obtain the rate of Ca^2+^ uptake (as shown in Figure 7A). Rates from at least 3 independent experiments were than averaged for reporting. As this method is highly sensitive, we found that 3 time points are sufficient for reliable quantification of the initiate rate. We now supplement data from several individual experiments in Figure 7—figure supplement 1 to give the readers a better sense about the variation in these experiments, and have also revised Materials and methods to make these issues clear.

Reviewer #2:In the study entitled "MICU1 interacts with the conserved aspartate ring of MCU to mediate Ca^2+^ activation of the mitochondrial Ca^2+^ uniporter", Phillips et al. seek to understand the mechanism by which the gate-keeping proteins MICU1 and MICU2 confer their regulatory effect on the pore forming subunit MCU of the mitochondrial calcium uniporter. In the absence of structural information on the entire MCU complex, it remains unclear how these MICU proteins regulate the ion conducting pore subunit. Using a combination of pull-down assays and fluorescent calcium uptake assays, the authors identified two conserved arginines (R119 and R154 of the human ortholog) in MICU1 that interact with the highly conserved aspartate residue of the selectivity filter of MCU. This direct interaction allows MICU1 to inhibit the Ca^2+^ uptake function of MCU at low cellular calcium concentrations.Overall, this study was well done and it addresses a fundamentally important question about how MCU is regulated. Additionally, the body of literature points to MICU1 as an important regulatory component of the uniporter since loss of MICU1 leads to mitochondrial calcium overload from cells to animal models to humans. Thus, the manuscript being considered is highly relevant and appropriate for publication in eLife.Reviewer #3:

We fully appreciate the reviewer’s suggestion to rely more on quantitative assays, such as FRET or ITC, to probe the interaction between MCU and MICU1. We do eventually hope to obtain binding parameters using these assays, but this would require purification of high-quality MCU proteins from higher eukaryotes, a challenging task that has not been achieved in the field. (MCU structures were determined recently, but these are homologues in fungi, which have no MICU1. Some of these fungal MCUs also show no function. An NMR structure published in 2016 used *C. elegans* MCU, but the protein was extracted from inclusion bodies in Fos-Choline- 14, a harsh detergent rarely used in membrane-protein biochemistry).

CoIP indeed has its limitation, but it also has unique advantages: the uniporter complex is properly assembled by cellular machineries in mitochondria, and the function of the complex can be assessed in native environments using Ca^2+^ flux assays. Importantly, the ability to substitute native proteins with point mutants using CRISPR/Cas9 now enables the detection of highly- specific protein-protein interactions. Our goal is to take full advantage of these strengths to address important questions in uniporter mechanisms. Below we provide a point-to-point response to explain why our conclusion regarding Ca^2+^ activation of the uniporter is within the resolution limit of our assays. We separate the reviewer’s comments into 3 parts, about issues related to the Ca^2+^ flux assay, CoIP, and writing.

Ca^2+^ flux assay:

In contrast, in the current manuscript, as I previously stated, the authors draw very specific mechanistic interpretations based on this yes/no assay. For example:Mutation of the D in the DIME signature sequence region of MCU to an A did not abolish Ca import but lead to loss of MICU1 interaction (as measured with co-IP). Furthermore, addition of 100 mM Na during the assay did not change the rate of Ca-import. The authors conclude that the D does not contribute to channel selectivity but mediates MICU1 interaction. The problem are the following: first, the assay does not address selectivity at all, but rather whether the mutation kills Ca transport. The authors should consider dropping any mention of the role of D in selectivity. Second, there should be a control experiment showing the effect of Na addition on WT MCU. Third, the finding that mutation of the D to any other amino acid (except E and A) abolished Ca-import is intriguing, and argues that the story is much more complex. Fourth, the authors should also present the coIPs with the D261E MCU mutant, which, according to the authors' hypothesis, is functional and expected to support the interaction with MICU1.

The reviewer is concerned that the fluorescence-based Ca^2+^ flux assay is not quantitative, and is suitable mostly for yes/no questions. Indeed, as in our response to reviewer #1, we strictly limit the use this assay for addressing yes/no questions. For instance, we show that D261A is Ca^2+^-transport competent, and that the transport is unaffected by adding 100 mM Na^+^ or Ru360. These are all valid conclusions well within the resolution of the assay.

As for the selectivity, the Clapham lab has shown that MCU employs a classical multi-ion pore mechanism in which Na^+^ can rapidly permeate the channel in the absence of Ca^2+^ and adding Ca^2+^ blocks the Na^+^ flux and leads to Ca^2+^ permeation. Therefore, if D261A significantly reduces MCU’s Ca^2+^ selectivity, adding 100 mM Na^+^ should strongly suppress the Ca^2+^ (only 10 µM) flux. That being said, a rigorous test of selectivity indeed requires electrophysiological experiments not performed in this work. We therefore decide to change the wording “selectivity” to “permeation” to be more accurate. As requested by the reviewer, we have performed experiments adding Na^+^ to WT MCU, and showed that Na^+^ has no effect on Ca^2+^ transport (Figure 3—figure supplement 1).

It’s true that several D261 mutations are non-functional, but this is not surprising as D261 sits in a critical position in the pore, right above the high-affinity Ca^2+^ site formed by E264. Mutations of D261 could in many non-specific ways perturb the chemistry required for Ca^2+^ transport. Finally, the original manuscript did provide the D261E CoIP data—it binds to MICU1 (please see Figure 4).

The conclusion that the Asp in the DIME sequence does not contribute to the channel's selectivity filter is premature as it is based on only one mutant that for some reason still supports Ca uptake and is not affected by Na addition in these particular assays. However, there is no control showing how Na affects these fluorescent assays in the WT or in any of the other mutants, and furthermore, the assay is still indirect.

We now change the sentence in the Results to “these results demonstrate that the DIME-Asp mediates MCU interaction with MICU1, instead of contributing essentially to MCU’s Ca^2+^ permeation.” The reviewer is concerned that this argument is based on a single D261A mutation. However, this is a very powerful positive result. Enzymatic reactions are known to require very specific chemistry. If the D261 side-chain is necessary for high-throughput Ca^2+^ permeation, it is extremely unlikely that other protein components can somehow compensate for a drastic change of the D261 side-chain to produce Ca^2+^ transport. That’s why we conclude that D261 does not contribute essentially to Ca^2+^ transport. It’s true that several other D261 mutants are non-functional, but again, such negative results are not particularly surprising, considering the critical position of the D261 residue in the pore. The assay is not quantitative, but it is direct and highly specific—it is a standard assay widely used to directly detect the uniporter’s transport activity. Our conclusion is derived well within the ability of the assay.

In Figure 6-7, the authors test whether MICU1 interaction with the pore in resting Ca concentrations is necessary to keep the pore closed. They find by coIP that the MICU1-MCU interaction is disrupted in the presence of 10 μm Ca and that in MICU1 knockout cells, Ca fluxes inside mitochondria, while if MICU1 is added back, less Ca is uptaken. If the MICU1 R mutants are added back instead, except for one of them, they are less competent than WT in restoring Ca flux. It would have been useful for the authors to also show that, similar to the R mutants on MICU1, the MCU D to A mutant (the interaction partner of the Rs) is also less competent than WT to restore Ca flux.

We have now performed ^45^Ca^2+^ flux experiments using MCU-KO cells transfected with WT or D261A MCU. Figure 7B shows that in submicromolar Ca^2+^, D261A produces mitochondrial Ca^2+^ uptake, while WT is completely inactive. This provides strong support for our model that MICU1 must bind to MCU to shut the uniporter in resting cellular conditions. While carrying out these experiments, we noticed that the rate of D261A-mediated Ca^2+^ “leak” is 6.2-fold slower than that observed using MICU1-KO cells. We investigated, and identified a few factors that might underlie the small magnitude of the leak, including a slower turnover rate of D261A than WT MCU, and other residues (e.g., S259) being involved in MICU1 binding. These are now described in the revised Result section.

CoIP related issues:

This entire study is predicated on the finding that MICU1 and MCU specifically interact directly and not through EMRE. This is shown exclusively with co IPs, which are usually only preliminary indicators of specific binding. It would be helpful for the solidity of the argument and for all the experiments that follow, for this interaction to be shown with more direct binding assays.

We share the reviewer’s concerns regarding the limitation of CoIP. Therefore, we endeavored to obtain multiple lines of evidence before drawing conclusions. For the MCU-MICU1 interaction, we show that it is highly-conserved so that human MICU1 can pull down MCU in lower eukaryotes (*i.e.* plants and protists), whose uniporters contain only MCU and MICU1 (no EMRE).

Moreover, this MCU-MICU1 interaction can be manipulated by point mutations: D261A at the cytoplasmic surface of MCU disrupts the complex, while E264A deeper in the pore does not affect MICU1 binding. This is analogous to the classical approach of utilizing point-directed mutagenesis to identify specific protein-protein interactions in binding assays.

The authors claim that the MCU-MICU1 interaction is electrostatic as evidenced by weaker interactions as measured by lower ratios of MCU/MICU1 gel bands post coIPs in higher ionic strength buffer conditions. If the authors want to make this electrostatic claim, they should show both a positive and a negative control experiment. Otherwise, a direct binding interaction assay would be more appropriate in this case.

The finding that the stability of the MCU-MICU1 complex can be manipulated by varying the ionic strength is diagnostic of electrostatic interactions (Figure 5A). This together with the observation that D261A and D261Q, but not D261E, abolishes MICU1 binding (Figure 4) provide strong evidence that the MCU-MICU1 interaction is electrostatic. The reviewer has a good point that controls should be provided. We now present data showing that the MICU1-2 interaction and the epitope interaction between 1D4-tagged MCU and the anti-1D4 antibody are unaffected by the ionic strength (Figure 5—figure supplement 1).

The authors screen for interacting partners for the Asp on MCU by mutating 18 conserved Arg and Lys residues on MICU1 to Ala and checking for interaction using coIPs. They find two Arg that disrupt MCU association and claim that these Rs on MICU1 mediate electrostatic interactions with the D on MCU. Perhaps these claims should be reduced to: the two Rs are important in mediating the interaction between MCU and MICU1, since there is no indication that the Rs interact with the D specifically. Furthermore, if the screen were not limited to positively charged residues, perhaps other residues would pop up as important to this interaction.

We completely agree with the reviewer that our data do not demonstrate directly that these two Rs interact with D261. Instead, the results indicate that these residues are crucial for the MCU- MICU1 interaction. We have been careful to make this clear, and have further revised the manuscript to emphasize this point.

The authors conclude that the MICU1 interaction with MCU must work by MICU1 directly blocking the MCU pore, predicated on their conclusions that the Rs on MICU1 electrostatically interact with the Ds in MCU, which are located in the pore. Since there is no solid evidence for the Rs interacting with the D, the pore blocking interpretation is similarly weak and I suggest that it is removed.

We disagree with this comment. An electrostatic interaction of MICU1 with D261 at the entrance of the MCU pore, whether it’s directly mediated by R119/R154 or not, would prevent Ca^2+^ entry into the pore, as in the classical example of charybdotoxin block of K channels. This is further confirmed by the quantitative functional analysis showing that mutations that disrupt MCU- MICU1 interaction compromise MICU1’s ability to close MCU. These data argue strongly that MICU1 shuts MCU in resting cellular conditions by blocking the pore.

These experiments appear well-executed but they fall short of demonstrating essentially any of the authors' claims on the MCU mechanism. The obvious components of the mechanism not demonstrated here are: MICU1 as MCU pore blocker, MICU1 unblocking the pore upon Ca- binding, the location of Ca binding for this mechanism, the involvement of the D in the DIME sequence in the interaction between the two proteins, how does Ca break the Asp-Arg mediated interaction between the two proteins, etc.

We hope that we have explained clearly why our results provide strong evidence that MICU1 closes MCU by electrostatically interact with the DIME-Asp to block the pore. We are unable to answer all questions in a single paper. As already stated in the last paragraph of our Discussion, future work is needed to test the role of MICU1’s individual EF hand in this mechanism, and to understand how Ca^2+^ breaks the MCU-MICU1 complex. The later most likely requires an atomic structure of the MCU-MICU1 subcomplex.

Comments on manuscript writing:

The Introduction is too short and lacks presentation of the selectivity of the channel (which is actually brought up in the results), as well as even a brief discussion of the existing structure(s). Despite the precision of the conclusions drawn regarding the mechanism of Ca-induced activation of MCU, the authors do not discuss this mechanism from the perspective of the structures, which is I believe necessary.

As the selectivity of the channel is not the main issue in this manuscript, we feel that discussing this in the Introduction would distract the readers from the main question of how MICU1 gates MCU. We did describe the classical multi-ion pore mechanism underlying Ca^2+^-channel selectivity in the Discussion. The Ca^2+^-activation mechanism of MCU is mediated by the MICU1 protein. Currently, there is no MCU-MICU1 subcomplex structure. Structures of MCU homologues from fungal species reveal MCU’s oligomer state and a possible Ca^2+^-selectivity mechanism, but provide very little insights into the mechanism by which MICU1 regulates MCU. (These fungal species also do not have MICU1). There are potential issues with the MICU1 structure as described in our response to reviewer #2. Thus, we feel that it’s premature to discuss these structures in the Introduction.

I am assuming that the radioactive Ca flux assay was used instead of the fluorescent one for Figure 6 because of the low Ca involved. However, this must be spelled out in the manuscript, because it looked like the fluorescent assay was quite sensitive. Furthermore, more information is needed for both the fluorescent assay and the radioactive assay (the section in the Materials and methods is not detailed enough). For instance, for the radioactive assays, I don't get a sense of what is plotted in Figure 6. Is this only one experiment, since there are no error bars? What's the signal to noise here? What do the signals look like before and after application of the Ru360? Etc

Please see our response to reviewer #1’s major concerns. We have extensively revised Materials and methods to provide more detailed information, and have also included the data before and after adding Ru360 in Figure 7—figure supplement 1.

Overall, for the reasons I outlined above, I believe that the authors overreach in their conclusions and interpretations. A few examples: Introduction last paragraph: "These results led to a molecular mechanism in which MICUs open or close the uniporter in response to intracellular Ca signals by physically blocking or unblocking the MCU pore."; first paragraph of Discussion: "Here, we establish a mechanism in which MICU1 shuts the uniporter by binding to the DIME-Asp side chain carboxylate ring to block the IMS entrance of the MCU pore..… Ca-binding to MICU1 at its EF hands disrupts this interaction, thus leading to opening of this Ca-activated Ca channel". Last two paragraphs of the Discussion: ".… a result that demonstrated unambiguously that S1 is not critical for Ca selectivity…..". "..the current study provides insight into the uniporter's ion selectivity mechanisms…". All these statements need to be rethought.

We hope that we have explained clearly that our conclusions are based on strong experimental supports. The word “selectivity” has been changed to “permeation.”